# Role of Syndecan-4 in the Inhibition of Articular Cartilage Degeneration in Osteoarthritis

**DOI:** 10.3390/biomedicines11082257

**Published:** 2023-08-12

**Authors:** Yoshio Hattori, Masahiro Hasegawa, Takahiro Iino, Kyoko Imanaka-Yoshida, Akihiro Sudo

**Affiliations:** 1Department of Orthopaedic Surgery, Mie University Graduate School of Medicine, Tsu 514-8507, Japan; yoshio418kingdom@yahoo.co.jp (Y.H.); tiino@med.mie-u.ac.jp (T.I.); a-sudou@clin.medic.mie-u.ac.jp (A.S.); 2Departments of Pathology & Matrix Biology, Mie University Graduate School of Medicine, Tsu 514-8507, Japan; imanaka@doc.medic.mie-u.ac.jp

**Keywords:** articular cartilage, osteoarthritis, syndecan-4, heparan sulfate proteoglycan

## Abstract

Despite its widespread existence, there are relatively few drugs that can inhibit the progression of osteoarthritis (OA). Syndecan-4 (SDC4) is a transmembrane heparan sulfate proteoglycan that modulates cellular interactions with the extracellular matrix. Upregulated SDC4 expression in articular cartilage chondrocytes correlates with OA progression. In the present study, we treated osteoarthritic cartilage with SDC4 to elucidate its role in the disease’s pathology. In this in vitro study, we used real-time polymerase chain reaction (PCR) to investigate the effects of SDC4 on anabolic and catabolic factors in cultured chondrocytes. In the in vivo study, we investigated the effect of intra-articular injection of SDC4 into the knee joints of an OA mouse model. In vitro, SDC4 upregulated the expression of tissue inhibitor of metalloproteinase (TIMP)-3 and downregulated the expression of matrix metalloproteinase (MMP)-13 and disintegrin and metalloproteinase with thrombospondin motifs (ADAMTS)-5 in chondrocytes. Injection of SDC4 into the knee joints of OA model mice prevented articular cartilage degeneration 6 and 8 weeks postoperatively. Immunohistochemical analysis 8 weeks after SDC4 injection into the knee joint revealed decreased ADAMTS-5 expression and increased TIMP-3 expression. The results of this study suggest that the treatment of osteoarthritic articular cartilage with SDC4 inhibits cartilage degeneration.

## 1. Introduction

Knee osteoarthritis (OA) is the most frequent cause of knee pain in individuals over the age of 50 years, and the risk of developing OA increases with age. One study stated that OA affects up to 30% of people >65 years of age, with another estimating that 10% of males and 17% of females over the age of 65 are affected [1,2]. The current treatment options for knee OA are as follows: conservative treatment; non-pharmacologic therapies, such as weight management through diet and exercise, unloader knee braces, and physical strengthening exercises; and pharmacologic therapy centered on NSAIDs. Possible surgical treatments include osteotomy and knee arthroplasty. OA is a whole joint disease involving all joint tissues: cartilage, meniscus, synovial membrane, infrapatellar fat pad, and subchondral bone [3,4]. Although OA is one of the most common and well-studied knee diseases, its pathophysiology remains poorly understood [5]. 

Degradative enzymes, also known as matrix metalloproteinases (MMPs), are upregulated in OA, triggering an imbalance that leads to the loss of proteoglycans and collagen. During OA development, chondrocytes tend to increase proteoglycan synthesis and produce tissue inhibitory factors for MMPs (TIMPs) to balance degradation, with restorative actions being inadequate to counteract these changes [5]. This imbalance leads to a decrease in proteoglycan content despite an increase in synthesis, higher water content, disrupted collagen structure, and reduced articular cartilage elasticity. These changes cause wear on the joint surfaces [6]. In addition, the articular cartilage has a poor self-repair capability; therefore, OA can develop due to increases in cartilage degeneration processes. Although numerous studies have sought to identify drugs that inhibit the progression of cartilage degeneration, few effective drugs are currently available [7].

Syndecan-4 (SDC4) is a ubiquitously expressed transmembrane heparan sulfate proteoglycan that modulates interactions with the extracellular matrix (ECM) [8]. Structurally, SDC4 comprises a short cytoplasmic domain, transmembrane domain, and extracellular domain (ectodomain) [9]. The SDC4 ectodomain interacts with multiple ECM molecules, growth factors, and cytokines via heparan sulfate chains [10]. During tissue repair, cells that mediate wound healing exhibit transient upregulation of SDC4 expression in conjunction with integrins. SDC4 further regulates mesenchymal cell function during tissue repair [11] and the activation of protein kinase C [12], focal adhesion kinase [13], and RhoA8 [14]. SDC4 expression is also upregulated in osteoarthritic chondrocytes of the articular cartilage, and this upregulation is correlated with OA progression [15]. In the hypoxic environment of the healthy intervertebral disc, SDC4 plays an important role in regulating the homeostasis of the medullary nuclei [16]. These diverse roles indicate that SDC4 plays a key role in cartilage biology. However, the role of SDC4 in OA pathogenesis and cartilage repair remains poorly understood.

The ectodomain of the SDC is cleaved and shed as a soluble proteoglycan when the core protein undergoes proteolysis. The ectodomain of shed SDC retains the binding properties of its cell surface precursors [17]. In wounds, the ectodomains of SDC1 and SDC4 are released into the exudate and modulate growth factor activity [18]. Increased levels of shed SDC4 have been observed in the synovial fluid of OA patients [19], where it maintains the balance between proteolysis and growth factor expression [20]. Shed SDC4 may further function in host defense during tissue repair [17]. Based on these observations, we hypothesized that treatment with the SDC4 ectodomain may exert beneficial effects on osteoarthritic cartilage. 

Synovitis is an important feature in the OA process and is defined as inflammation of the synovium [21]. It may manifest itself phenotypically as a thickening of the synovial membrane or indirectly as joint effusion as the result of synovial activation [22]. One study demonstrated a positive correlation between the severity of synovitis and the degree of progression of cartilage lesions over time [23], suggesting that synovitis in OA predisposes to further structural progression [24,25]. The infrapatellar fat pad (IFP) is an intracapsular and extrasynovial adipose tissue structure in the knee joint that is closely associated with synovitis [26]. It is suggested that the IFP could be an important player in OA [27]. IFP could have both protective and disease-enhancing effects in OA [26]. In light of previous reports that identified shed SDC4 in the joint fluid of patients with OA [19] and that SDC4 is associated with OA progression [15], the possibility that shed SDC4 induces synovitis must also be considered.

Based on the existing literature, this study aimed to investigate the effects of the SDC4 ectodomain treatment on osteoarthritic cartilage. Specifically, we investigated the in vitro effects of SDC4 on anabolic and catabolic factors in cultured chondrocytes and examined the effect of intra-articular injection of SDC4 in an in vivo study using an OA mouse model. This study presents new findings that investigate the role of the SDC4 extracellular domain in articular cartilage. 

## 2. Materials and Methods

### 2.1. Experimental Animal Models

Ninety male 8-week-old BALB/c mice weighing approximately 22 g (SLC, Hamamatsu, Japan) were used as models in this study. The mice were maintained in accordance with the ARRIVE guidelines. The study protocol was approved by the Institutional Ethics Review Board (Department of Mie University Medical and Hospital Management; approval number 2019-40).

### 2.2. Intra-Articular Injection of SDC4 in OA Model Mice

To evaluate the effect of SDC4 on the cartilage, we microscopically analyzed mouse knee joints. A surgical procedure was performed on mice to create an experimental OA model. The mice were anesthetized via subcutaneous injection of medetomidine hydrochloride (0.75 mg/g body weight), midazolam (4 mg/g body weight), and butorphanol tartrate (5 mg/g body weight). One knee joint was exposed via a medial parapatellar incision. After dissecting the anterior cruciate and medial collateral ligaments, the joint capsule and skin were separately closed. Subsequently, the articular capsule was closed, and then mice in the SDC4 group (*n* = 45) received an injection of 1 μg/mL recombinant human SDC4 (10 μL) (rhSDC4, Glu19-Glu145; R&D Systems, Minneapolis, MN, USA, #2918-SD) into the knee using the Trance patella tendon approach. The gene sequence from which the recombinant protein used in this study was derived encodes the ectodomain of SDC4, while rhSDC4 used in the treatment of mice represents the extracellular domain. Mice in the control group (*n* = 45) received an injection of phosphate-buffered saline (PBS; 10 μL). The mice were randomly divided into groups in an alternate manner. After surgery, all the mice were able to walk freely without the need for a splint. The mice were kept in a laboratory animal facility (five mice per cage) at 24–25 °C, provided with standard mouse chow and water ad libitum, and maintained under a 12 h light–dark cycle.

### 2.3. Analysis of SDC4 Injected into Knee Joints

The HiLyte Fluor 555 labeling kit (do Labo, Kumamoto, Japan) was used to label rhSDC4. NH2-Reactive HiLyte Fluor 555 has a succinimidyl ester group and readily forms covalent bonds with amino groups on target proteins and other macromolecules without requiring activation. Small molecules, such as Tris buffer and amine compounds, that could interfere with the assay or labeling reaction were removed from the protein samples via filtration. Fluorescently labeled rhSDC4 (10 μL) was administered to one knee of a 15 OA model. Three mice were used for each time point. The entire knee joint was dissected at 1 and 4 days or at 1, 2, and 4 weeks postoperatively. The frozen sections were mounted on silane-coated glass slides and air-dried. Hoechst 33342 (Sigma-Aldrich, St. Louis, MO, USA) was used for nuclear staining for 5 min.

### 2.4. Histopathological Assessment

Mice were euthanized using CO_2_ at 2 weeks (control group, *n* = 9; SDC4 group, *n* = 9), 4 weeks (control group, *n* = 9; SDC4 group, *n* = 9), 6 weeks (control group, *n* = 9; SDC4 group, *n* = 9), 8 weeks (control group, *n* = 9; SDC4 group, *n* = 9), or 12 weeks (control group, *n* = 9; SDC4 group, *n* = 9). The entire knee joint was dissected. All samples were fixed in 4% formalin for 2 days at room temperature, demineralized with 10% ethylenediaminetetraacetic acid, dehydrated, embedded in paraffin, and sliced at 4 μm thickness.

### 2.5. Histopathological Evaluation

#### 2.5.1. Histological Grading of Cartilage and the Synovial Membrane

Sections were stained with hematoxylin, eosin, and safranin-O. All specimens were scored blindly by three independent investigators.

Synovial membrane: Synovitis in the synovial membrane was assessed using the synovitis score, which ranged from 0 to 9 points. The degree of synovitis was measured as the enlargement of the synovial lining cell layer on a scale of 0–3 (0 = one layer, 1 = 2–3 layers, 2 = 4–5 layers, and 3 = more than 5 layers), density of the resident cells on a scale of 0–3 (0 = normal cellularity, 1 = slightly increased cellularity, 2 = moderately increased cellularity, and 3 = greatly increased cellularity), and inflammatory cell infiltration on a scale of 0–3 (0 = no inflammatory cell infiltration, 1 = few infiltrating cells, 2 = numerous lymphocytes or plasma cells, and 3 = dense band-like inflammatory infiltration) [28]. We compared the synovitis scores of both groups at 2, 4, 6, 8, and 12 weeks post-injection.

Cartilage: Cartilage degeneration was assessed using Mankin [29] and OARSI scores [30]. The Mankin score was calculated as the sum of the scores in four categories of histological features and ranged from 0 to a maximum of 14 points: cartilage anatomy was graded from 0 (normal tissue) to 6 points (complete loss of cellular organization, clusters of cells, osteoclastic activity); cellular abnormality was graded from 0 (normal) to 3 points (hypocellularity); matrix staining (with safranin-O) was graded from 0 (normal or slightly diminished staining) to 4 points (non-staining); and tidemark integrity was graded from 0 (intact) to 1 point (destruction) [29]. The OARSI score ranges from 0 to a maximum of 6 points: a score of 0 represents normal cartilage; 0.5 = loss of proteoglycan with an intact surface; 1 = superficial fibrillation without loss of cartilage; 2 = vertical clefts down to the layer immediately bellow the superficial layer and some loss of surface lamina; 3 = vertical clefts/erosion of the calcified layer lesion for 1–25% of the quadrant width; 4 = lesion reaches the calcified cartilage for 25–50% of the quadrant width; 5 = lesion reaches the calcified cartilage for 50–75% of the quadrant width; and 6 = lesion reaches the calcified cartilage for >75% of the quadrant width [30]. Histological grading scores for cartilage degeneration were determined separately for the medial femoral condyle and medial tibial plateau. Scores were compared between the groups at 2, 4, 6, 8, and 12 weeks postoperatively, and the mean scores were reported.

#### 2.5.2. Immunohistochemistry

ADAMTS-5 immunostaining was performed as previously described [31]. Sections were incubated in methanol containing 0.3% H_2_O_2_ for 30 min to block intrinsic peroxidase activity and subsequently heated in sodium citrate (pH 6) at 95 to 100 °C for 15 min according to the heat-induced epitope retrieval (HIER) method. The sections were then overlaid with a primary antibody against ADAMTS-5 (1:100, rabbit polyclonal; Abcam, Cambridge, UK) and incubated overnight. After washing, sections were incubated with peroxidase-conjugated anti-rabbit IgG Fab’ (1:100 dilution; DAKO, Glostrup, Denmark) for 1 h at 37 °C. Finally, an immune reaction was developed using a diaminobenzidine/H_2_O_2_ solution.

TIMP-3 and MMP-13 immunostaining was performed using a standard technique (Histofine Simple Stain Mouse Stain Kit; Nichirei Co., Tokyo, Japan) to block intrinsic mouse immunoglobulin activity. Sections were incubated in methanol containing 0.3% H_2_O_2_ for 30 min to block intrinsic peroxidase activity, and antigen retrieval was performed using the HIER method. After washing, the sections were treated with Histofine blocking reagent A for 60 min at 37 °C, followed by overnight incubation with a primary antibody against TIMP-3 (1:100, mouse monoclonal antibody, Kyowa Pharma Chemical Co., Ltd., Tokyo, Japan) or MMP-13 (1:100, mouse monoclonal antibody, Kyowa Pharma Chemical Co., Ltd.) at 37 °C. After washing, the sections were treated with Histofine blocking reagent B for 10 min at 37 °C, washed again, and then incubated with Histofine simple stain mouse MAX-PO (Nichirei Co., Tokyo, Japan) for 60 min at 37 °C. Finally, color was developed using a diaminobenzidine/H_2_O_2_ solution.

The results are expressed as the percentage of cells that stained positive for the respective antigen (TIMP-3, ADAMTS-5, or MMP-13) in the cartilage, with a maximum value of 100%. For statistical purposes, data from all specimens (defined as the loaded region of the tibial plateau and the femoral condyle) were considered. The data presented are the averages of three fields [32].

### 2.6. Chondrocyte Isolation and Culture

Human cartilage specimens were obtained from the femoral condyles of 20 patients (Kellgren and Lawrence grade 3: 10 patients; grade 4: 10 patients) who underwent total knee joint replacement for OA treatment. All patients provided informed consent, and the study was approved by the local ethics committee (Department of Mie University Medical and Hospital Management, approval number H2020-235). Cartilage fragments damaged by OA change were excised from the femoral condyles of the knee joints using a sharp curette. Cartilage fragments were incubated in 0.8% Pronase solution (Calbiochem, Darmstadt, Germany) and dissolved in Dulbecco’s modified Eagle’s medium/Ham F12 (DMEM/F12) (Gibco, Grand Island, NY, USA) for 30 min at 37 °C, with continuous agitation in an atmosphere of 5% CO_2_. After washing with DMEM/F12, the cartilage pieces were incubated with 0.4% collagenase (Roche Diagnostics, Penzberg, Germany) in DMEM/F12 for 90 min at 37 °C with orbital mixing. The cell suspension was then filtered using a 70 μm pore size nylon filter (BD Biosciences, Bedford, MA, USA) to remove tissue debris. The filtrate was centrifuged for 5 min at 1200 rpm. The cells were washed three times with DMEM/F12 containing 10% fetal bovine serum (FBS) and plated on 96- or 6-well tissue culture plates (Becton Dickinson Labware, Franklin Lakes, NJ, USA) in DMEM/F12 supplemented with 10% FBS, amphotericin B solution 0.25 μg/mL (Sigma Chemical Co., St. Louis, MO, USA), kanamycin (110 g/mL (Gibco), penicillin–streptomycin (penicillin 100 IU/mL, streptomycin 100 μg/mL) (Gibco), and 25 μg/mL ascorbic acid (Sigma). Chondrocytes were grown at 37 °C in a humidified atmosphere containing 5% CO_2_ and 95% air.

### 2.7. RNA Extraction and cDNA Synthesis

Cells were seeded at 1×10^5^ cells/well in 6-well plates (Becton Dickinson Labware, Franklin Lakes, NJ, USA) and incubated for 7 days. After the cultured chondrocytes reached confluency, they were treated with 0 or 1 μg/mL of rhSDC4 with 0.1% bovine serum albumin. After exposure to SDC4 for 24 h, chondrocytes were collected, and mRNA was extracted. Total RNA was isolated using the RNeasy Plus Mini kit (QIAGEN, Hilden, Germany) according to the manufacturer’s instructions. Complementary DNA (cDNA) was synthesized using oligo(dT) 15 priming of 1 μg of total RNA, using a cDNA synthesis kit (Roche Diagnostics, Penzberg, Germany) according to the manufacturer’s protocol.

### 2.8. Real-Time PCR

mRNA expression in the SDC4-treated group (SDC4 group) was compared with that in the untreated group (control group) using real-time PCR. The expression of anabolic factors (fibroblast growth factor (bFGF) (*n* = 9), transforming growth factor–beta (TGFb) (*n* = 9), TIMP-3 (*n* = 9)) and catabolic factors (ADAMTS-4 (*n* = 8), ADAMTS-5 (*n* = 5), MMP-13 (*n* = 6)) was evaluated.

TaqMan gene expression assay primer–probe pairs were used to detect MMP-13 (assay ID: Hs0000233992-m1) (*n* = 6), ADAMTS-4 (assay ID: Hs00192708-m1) (*n* = 8), ADAMTS-5 (assay ID: Hs00199841-m1) (*n* = 5), TIMP-3 (assay ID: Hs00165949-m1) (*n* = 9), bFGF (assay ID: Hs00266645-m1) (*n* = 9), TGFb (assay ID: Hs00998133-m1) (*n* = 9), and glyceraldehyde-3-phosphate dehydrogenase (GAPDH) (assay ID: 4325792) (*n* = 9). Quantitative cDNA analysis was performed using an ABI Prism 7000 Sequence Detector System (Applied Biosystems, Foster City, CA, USA) and a TaqMan Fast Advanced Master Mix System (Applied Biosystems). The thermal cycling conditions were 50 °C for 2 min, 95 °C for 2 min, and 40 cycles of 95 °C for 3 s and 60 °C for 30 s. GAPDH was used as an internal control housekeeping gene. The fold change in the level of each mRNA (SDC4 group/control group) was normalized to that of GAPDH.

### 2.9. Statistical Analysis

Statistical significance was determined using the Mann–Whitney U test. All statistical analyses were performed using EZR (Saitama Medical Center, Jichi Medical University, Saitama, Japan), a graphical user interface for R (R Foundation for Statistical Computing, Vienna, Austria, version 3.5.2). More precisely, this software is a modified version of the R commander (version 2.5-1) designed to add statistical functions frequently used in biostatistics [33]. Statistical significance was set at *p*-value < 0.05.

## 3. Results

### 3.1. Animal Welfare

All mice resumed normal activity and weight bearing immediately after recovery from anesthesia. No complications such as joint contractures or infections were observed in any of the mice.

### 3.2. Distribution of SDC4 Injected into Knee Joints

To examine how the injected SDC4 was distributed to each of the target tissues, the cartilage injected with labeled rhSDC4 into the joint was evaluated by fluorescence microscopy at 1 and 4 days and 1, 2, and 4 weeks after injection. Red fluorescence was observed in the cartilage matrix and synovium of frozen sections one week after injection. Although the fluorescence of the articular cartilage was of low intensity 1 week after injection, exogenously administered SDC4 remained for at least 1 week after injection. In contrast, no red fluorescence was observed in the cartilage of the control mice (Figure 1).

### 3.3. Histological Analysis and Grading of OA Model Mice

To evaluate the effect of SDC4 on articular cartilage, isolated knee joints were analyzed microscopically.

Cartilage: At 2 weeks, the surface of the articular cartilage of mice in both groups was clearly and uniformly stained with safranin-O. At four weeks, mild proteoglycan loss was observed in the cartilage of both groups. At 6 and 8 weeks, the development of OA was reduced in the SDC4 group compared with that in the control group (Figure 2). Notable alterations in surface structure and proteoglycan loss were observed in the control group. In contrast, the articular cartilage in the SDC4-treated group showed less proteoglycan loss. Articular lesions were assessed on a scale of 0–14 using the Mankin score (Figure 3a) and on a scale of 0–6 using the OARSI grading (Figure 3b). The Mankin scores in the control group were significantly higher than those in the SDC4 group at 6 and 8 weeks (6 weeks: *p* = 0.035; 8 weeks: *p* = 0.003). The OARSI score was significantly higher in the SDC4 group at 6 and 8 weeks (6 weeks, *p* = 0.032; 8 weeks, *p* = 0.004). There were no significant differences between the groups in the Mankin and OARSI scores at 2, 4, or 12 weeks. These results indicate that greater progressive cartilage degeneration occurred in control mice at 6 and 8 weeks than in SDC4-treated mice. At 12 weeks, the development of OA was observed in both groups with no significant difference between the groups.

Synovium: At 2 and 4 weeks, cellularity increased in the synovium, and the synovial lining cell layer was enlarged in both groups. At 6 weeks, the increased cellularity and enlarged lining of the cell layer improved in both groups. Low-grade synovitis occurred at 2 and 4 weeks in both groups, but improved at 6 weeks (Figure 4). There were no significant differences in the average synovitis scores between the SDC4 and control groups at any time point (Figure 5). These changes were thought to be attributable to surgical intervention, and SDC4 treatment did not exacerbate synovitis.

### 3.4. Immunohistochemistrical Analysis 

The expression of ADAMTS-5 and TIMP-3 was examined in the cartilage of OA model mice in the SDC4 and control groups each week using immunohistochemical analysis (Figure 6). No significant staining was observed with the anti-MMP-13 antibody. In the SDC4 group at 8 weeks after treatment, the percentage of cells that stained positive for ADAMTS-5 was significantly lower than that in the control group (Figure 7a), whereas the percentage of cells that stained positive for TIMP-3 was significantly higher than that in the control group (Figure 7b) (ADAMTS-5: SDC4, 15%; control: 20%, *p* = 0.01; TIMP-3: SDC4, 30%, control: 22%, *p* = 0.03).

### 3.5. Gene Expression in Chondrocytes

The function of SDC4 in regulating the expression of catabolic and anabolic factors in human OA chondrocytes was investigated using real-time PCR (Figure 8). Regarding anabolic factors, treatment with SDC4 upregulated the expression of TIMP-3 (1.67-fold increase, *p* = 0.038). Regarding catabolic factors, treatment with SDC4 downregulated the expression of MMP-13 (0.21-fold downregulation, *p* = 0.01) and ADAMTS-5 (0.47-fold downregulation, *p* = 0.007). No significant differences were observed in the expression of other genes in cells treated with SDC4.

## 4. Discussion

In the present study, we investigated the effect of treatment with the ectodomain of SDC4 on articular cartilage to determine whether SDC4 has therapeutic potential in the treatment of OA. SDCs are transmembrane heparan sulfate proteoglycans with heparan sulfate chains in the ectodomain. Through the heparan sulfate chain, SDCs bind to various soluble and insoluble ligands, including ECM components, cell adhesion molecules, growth factors, cytokines, proteases and protease inhibitors, lipid metabolism proteins, and microbial pathogens [34,35]. The ectodomain of SDCs with heparan sulfate chains is shed intact following the proteolysis of the core protein. These shed SDCs retained their binding properties and could bind to the same ligands as cell-surface SDCs [17]. These ectodomains are found in the fluid produced after injury or inflammation [18]. Prior studies have reported that SDC4 expression increases in correlation with the severity of OA [15]. Consistent with this correlation, increased levels of shed SDC4 have been reported in the synovial fluid of patients with OA, and SDC4 shedding was shown to be correlated with OA severity [19]. These findings suggest that SDCs are shed at sites of inflammation and that SDC4 has some function in osteoarthritic cartilage. The rhSDC4 used in the present study is a protein with an ectodomain of the whole protein. Therefore, we investigated the function of shed SDC4 by treating the articular cartilage of OA model mice with the SDC4 ectodomain.

In vitro, SDC4 treatment upregulated the expression of TIMP-3 and downregulated MMP-13 and ADAMTS-5 expression in chondrocytes. The in vivo immunohistochemical staining confirmed the results of the in vitro study. The intra-articular injection of SDC4 increased the number of TIMP-3-positive cells and decreased the number of ADAMTS-5-positive cells in the cartilage of OA mice. However, no change in MMP-13 expression was observed immunohistochemically following intra-articular injection of SDC4. These results suggest that the treatment of articular cartilage with SDC4 inhibits the progression of cartilage degeneration.

Aggrecan is a major structural component of cartilage that forms very large aggregates with hyaluronic acid, which is confined within a network of collagen II. This complex structure provides compressive strength to the ECM of the cartilage, allowing articular cartilage to be shock-absorbent [36]. As aggrecan plays a role in preventing the loss of collagen fibers, a reduction in aggrecan levels is characteristic of early OA [37]. Thus, preventing a decline in aggrecan levels is important in the early management of OA.

MMPs, particularly MMP-13, degrade type II collagen, while ADAMTS-5 is involved in aggrecan degradation. TIMP-3 is an aggrecanase inhibitor that functions as an endogenous inhibitor of these catabolic factors [36]. Studies in mice lacking TIMP-3 showed spontaneously elevated MMP activity and age-related cartilage degeneration, as observed in patients with OA [38]. TIMP-3 is the only member of the TIMP family capable of inhibiting both ADAMT and MMPs [39]. Stable expression of aggrecan-degrading enzymes and TIMP-3 in the articular cartilage maintains aggrecan homeostasis. When this balance is disturbed, aggrecan loss occurs, leading to OA development [40]. The present study demonstrated that the injection of SDC4 into the knee joint prevents articular cartilage degeneration in mice. In addition, labeled SDC4 injected into the joint remained in the cartilage matrix and synovium one week after injection. These findings suggest that SDC4 acts on the surface of the articular cartilage and may protect it from proteoglycan loss.

Several prior studies have shown that inactivation of SDC4 inhibits cartilage degeneration [13,32]. In particular, Echtermeyer et al. showed that intra-articular injection of an anti-SDC4 antibody reduced MMP-3 expression and ADAMTS-5 activity in chondrocytes, resulting in reduced cartilage degradation. The number of SDC4-expressing chondrocytes correlates with the degree of typical osteoarthritic changes and the histological severity of OA [15].

The role of the SDC4 ectodomain, which is shed from the cell surface at sites of inflammation, remains unclear. Therefore, we treated the articular cartilage with the extracellular domain of SDC4 and investigated its characteristics. The results of the present study were different from those of previous studies that examined the role of SDC4 [15,41]. Injection of the SDC4 ectodomain into the knee joints of OA mice prevented articular cartilage degeneration and inhibited ADAMTS-5 expression. In addition, intra-articular injection of SDC4 did not induce joint inflammation in model mice. These findings suggest that SDC4 is a transmembrane proteoglycan and that the SDC4 ectodomain functions differently. The ectodomain of shed SDC4 retains the binding properties of its cell surface precursors [17]. It has further been reported that levels of the SDC4 ectodomain are increased in the synovial fluid of patients with OA and that SDC4 shedding is correlated with OA severity [19]. Furthermore, shed SDC4 plays a role in the host defense during tissue repair [17]. In light of the above, these findings suggest that in OA, a pathological condition involving complex mechanisms, SDC4 exerts dual functions depending on whether it is present on the cell surface or shed. SDC4 promotes OA progression or inhibits the progression of cartilage degeneration as a self-protective response. In the present study, administration of the SDC4 ectodomain to the articular cartilage upregulated anabolic factors and downregulated catabolic factors, indicating that administration of the SDC4 ectodomain is effective at preventing cartilage degeneration. This is a novel finding in the investigation of SDC4, which has complex functions, and suggests that SDC4 may be useful in preventing the onset of OA. However, the mechanisms underlying these results remain to be elucidated, and further studies are needed to determine whether the treatment of articular cartilage with SDC4 is an effective therapeutic strategy for OA.

The present study had some limitations. First, the sample size was small. Second, we were unable to directly demonstrate that the intra-articular injection of SDC4 remained intact and physiologically active over time. Third, we did not determine which intracellular signaling pathways played a role in the present results. Elucidating the signaling pathway through which the extracellular domain of SDC4 inhibits articular cartilage degeneration and exerts other effects of SDC4 on articular cartilage may help in the development of therapies that inhibit OA progression.

## 5. Conclusions

Overall, in the present study, we investigated the role of the ectodomain of SDC4 in articular cartilage. In an in vivo study, SDC4 inhibited cartilage degeneration without synovitis in an OA mouse model at 6 and 8 weeks. In an OA mouse model, SDC4 treatment decreased ADAMTS-5 expression and increased TIMP-3 expression in chondrocytes after 8 weeks. In vitro, SDC4 treatment upregulated the expression of TIMP-3 and downregulated MMP-13 and ADAMTS-5 expression in chondrocytes. The results of this study suggest that the treatment of articular cartilage with the ectodomain of SDC4 may have an inhibitory effect on cartilage degeneration. Further studies are required to determine whether treatment with the ectodomain of SDC4 is a viable therapeutic approach for the treatment of OA.

## Figures and Tables

**Figure 1 biomedicines-11-02257-f001:**
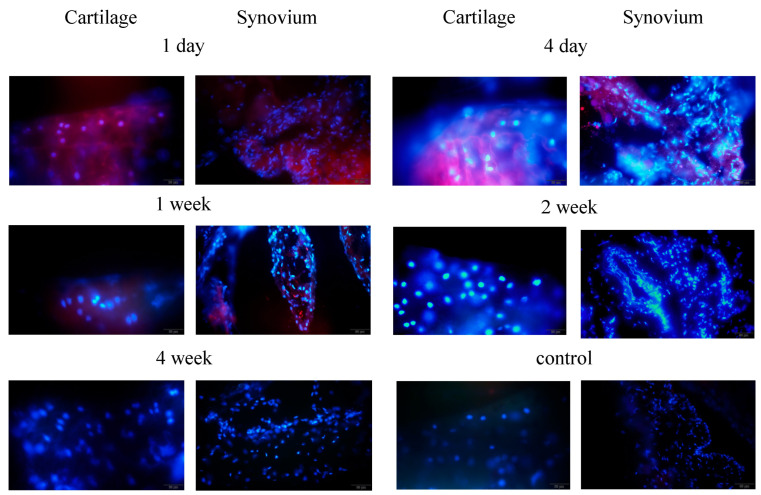
Distribution of injected SDC4 (SDC4: red) in the cartilage and synovium. Nuclei are labeled in blue. (Scale bar: Cartilage 20 μm, Synovium 50 μm).

**Figure 2 biomedicines-11-02257-f002:**
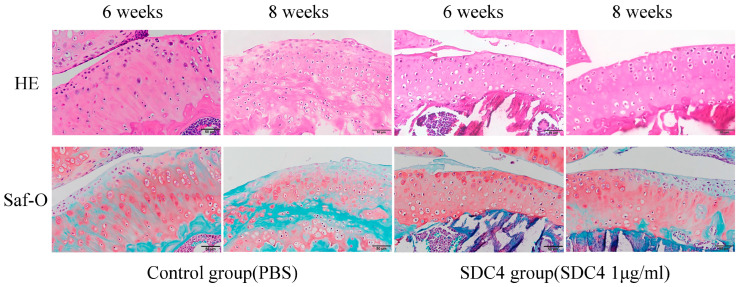
Histologic analysis of surgically induced OA in the cartilage tissue of the knee joints of mice following treatment with SDC4 or phosphate-buffered saline (PBS) control; hematoxylin and eosin (H&E) and safranin-O staining. (Scale bar: 50 μm).

**Figure 3 biomedicines-11-02257-f003:**
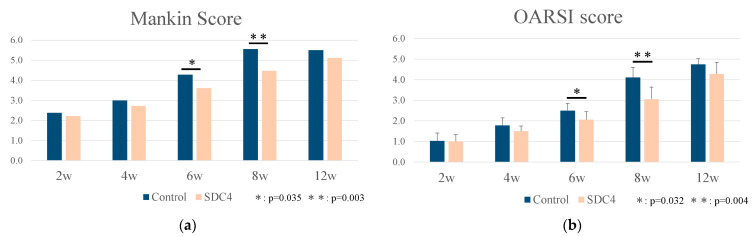
(**a**) Mankin Score; (**b**) OARSI score. All parameters were taken as counts of the cartilage tissue.

**Figure 4 biomedicines-11-02257-f004:**
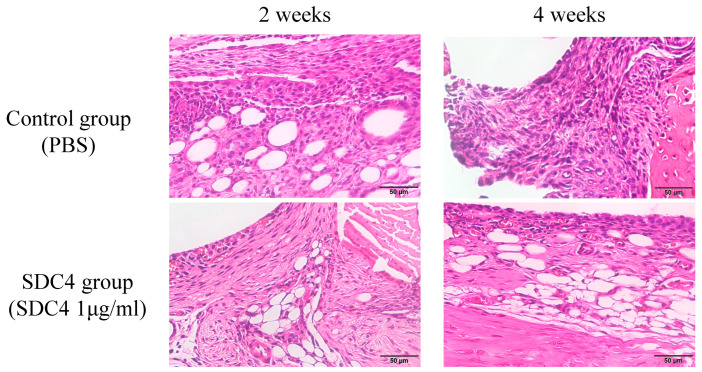
Histologic analysis of synovitis in the synovium tissue of the knee joints of mice after treatment with SDC4 or PBS control. H&E staining. (Scale bar: 50 μm).

**Figure 5 biomedicines-11-02257-f005:**
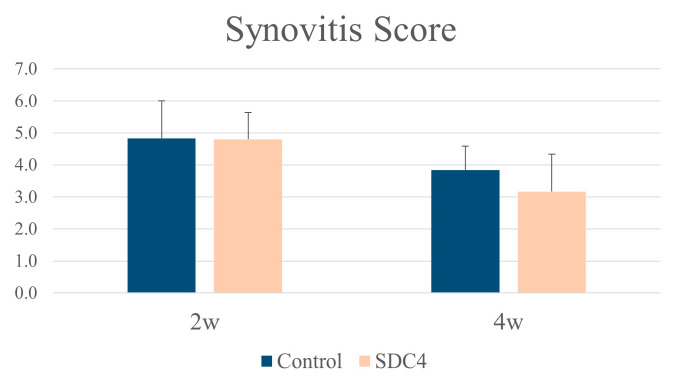
Synovitis scores in both groups at 2 and 4 weeks. All parameters were taken as counts of the synovium tissue.

**Figure 6 biomedicines-11-02257-f006:**
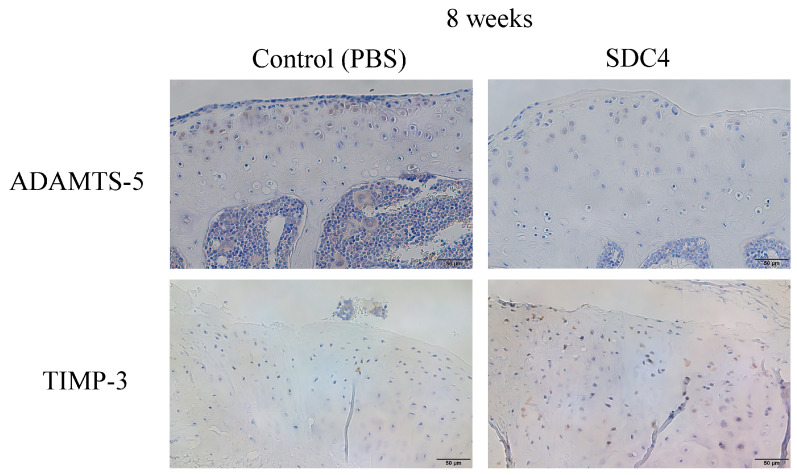
Immunohistochemical analysis of the cartilage tissue. (Scale bar: 50 μm).

**Figure 7 biomedicines-11-02257-f007:**
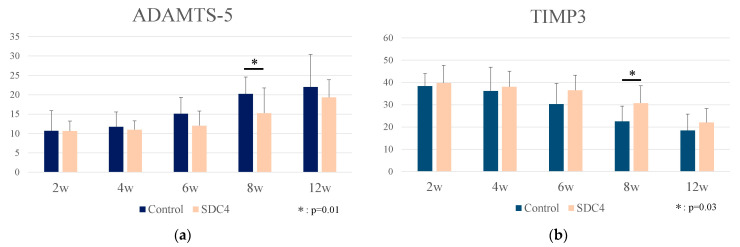
(**a**) Percentage of ADAMTS-5-positive chondrocytes as determined by immunostaining. (**b**) Percentage of TIMP-3-positive chondrocytes as determined by immunostaining. All parameters were taken as counts of the cartilage tissue.

**Figure 8 biomedicines-11-02257-f008:**
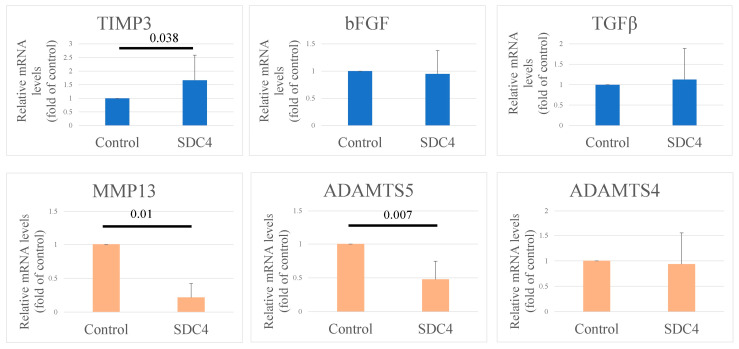
Analysis of mRNA expression in cultured cells. MMP-13: *n* = 6, ADAMTS-4: *n* = 8, ADAMTS-5: *n* = 5, TIMP-3: *n* = 9, bFGF: *n* = 9, TGFb: *n* = 9, and glyceraldehyde-3-phosphate dehydrogenase (GAPDH): *n* = 9.

## Data Availability

All original data of the study are presented in the article, and all relevant queries can be directed to the corresponding author.

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
