# Peer review of "Role of Syndecan-4 in the Inhibition of Articular Cartilage Degeneration in Osteoarthritis"

_biomedicines, 2023, doi:10.3390/biomedicines11082257_

Round 1

Reviewer 1 Report

This manuscript entitled “Role of Syndecan-4 in Inhibiting Articular Cartilage Degeneration in Osteoarthritis”, studies the effect of Syndecan-4 (SDC4) on cartilage degeneration of osteoarthritis (OA) through in vitro cultured chondrocytes and in vivo mice OA model in which histopathological evaluation and expressions of anabolic factors of beta-fibroblast growth factor, transforming growth factor of beta (TGFb) and catabolic factors of ADAMTS-4, ADAMTS-5, and MMP-13 using QPCR were analyzed. Results showed SDC4 treatment could upregulate TIMP-3 expression and downregulate MMP-13, disintegrins and ADAMTS expressions in chondrocytes; in addition, in vivo mice model also showed SDC4 treatment could prevent against articular cartilage degeneration.

Overall, this study provides more valuable data for academic and clinical reference. However, the text is hard to read that may be due to language limitation, it should be revised by a English-native expert. Additionally, there are still some need to be clarified as the following:

1. The specimen source (such as culture cells or tissue) and numbers should be descripted in the figure legends.

2. Does line 309 stating “investigated in human OA chondrocytes using real-time PCR (Fig. 10) correct?

3. The efficacy and mechanisms of SDC4 treatment on OA either found in this study or other studies should be discussed deeply since it is the core of this manuscript.

Author Response

To Reviewer 1

This manuscript entitled “Role of Syndecan-4 in Inhibiting Articular Cartilage Degeneration in Osteoarthritis”, studies the effect of Syndecan-4 (SDC4) on cartilage degeneration of osteoarthritis (OA) through in vitro cultured chondrocytes and in vivo mice OA model in which histopathological evaluation and expressions of anabolic factors of beta-fibroblast growth factor, transforming growth factor of beta (TGFb) and catabolic factors of ADAMTS-4, ADAMTS-5, and MMP-13 using QPCR were analyzed. Results showed SDC4 treatment could upregulate TIMP-3 expression and downregulate MMP-13, disintegrins and ADAMTS expressions in chondrocytes; in addition, in vivo mice model also showed SDC4 treatment could prevent against articular cartilage degeneration.

Thank you very much for kind review on our manuscript.

Overall, this study provides more valuable data for academic and clinical reference. However, the text is hard to read that may be due to language limitation, it should be revised by an English-native expert.

According to your suggestion, the manuscript was revised by an English-native expert.

Additionally, there are still some need to be clarified as the following:

  1. The specimen source (such as culture cells or tissue) and numbers should be descripted in the figure legends.

According to your suggestion, we added the specimen source and numbers in the figure legends as follows.

Figure 1. Distribution of injected SDC4(SDC4: red) in the cartilage and synovium. Nuclei are labeled in blue. (Lines 252)

Figure 2. Histologic analysis of surgically-induced OA in the cartilage tissue of the knee joints of mice following treatment with SDC4 or phosphate-buffered saline (PBS) control; hematoxylin and eosin (H&E) and safranin-O staining.  (Lines 274-275)

Figure 3. (a) Mankin Score, (b) OARSI score. All parameters were taken as counts of the cartilage tissue. (Lines 279)

Figure 4. Histologic analysis of synovitis in the synovium tissue of the knee joints of mice after treatment with SDC4 or PBS control. H&E staining. (Lines 289-290)

Figure 5. Synovitis Scores in both groups at 2 and 4 weeks. All parameters were taken as counts of the synovium tissue. (Lines 294)

Figure 6. Immunohistochemical analysis of the cartilage tissue (Line 306)

Figure 7. (a) Percentage of ADAMTS-5–positive chondrocytes as determined by immunostaining. (b) Percentage of TIMP-3–positive chondrocytes as determined by imunostaining. All parameters were taken as counts of the cartilage tissue. (Lines 311-313)

Figure 8. Analysis of mRNA expression in cultured cells.  MMP-13: n=6, ADAMTS-4: n=8, ADAMTS-5: n=5, TIMP-3: n=9, bFGF: n=9, TGFb: n=9, and glyceralde-hyde-3-phosphate dehydrogenase (GAPDH): n=9 (Lines 324-326)

  1. Does line 309 stating “investigated in human OA chondrocytes using real-time PCR (Fig. 10) correct?

According to your kind comment, we have made the following correction.

  Fig. 10 →Fig. 8 (Line 316)

  1. The efficacy and mechanisms of SDC4 treatment on OA either found in this study or other studies should be discussed deeply since it is the core of this manuscript.

According to your suggestion, we added to the discussion our considerations in SDC4 treatment, especially what we would like to emphasize from the results of this study as follows.

. In the present study, administration of the SDC4 ectodomain to the articular cartilage upregulated anabolic factors and downregulated catabolic factors, indicating that ad-ministration of the SDC4 ectodomain is effective at preventing cartilage degeneration. This is a novel finding in the investigation of SDC4, which has complex functions, and suggests that SDC4 may be useful in preventing the onset of OA. However, the mechanisms underlying these results remain to be elucidated, and further studies are needed to determine whether treatment of articular cartilage with SDC4 is an effective therapeutic strategy for OA. (Lines 394-401)

Reviewer 2 Report

The manuscript by Hattori, et al. entitled: “ Role of Syndecan-4 in Inhibiting Articular Cartilage 2 Degeneration in Osteoarthritis” (biomedicines-2492434) presents a study on the effects of SDC4 ectodomain treatments on articular cartilage degradation in surgically induced OA in mice and expression of the selected catabolic and anabolic factors in human OA chondrocytes in vitro. The authors found that SDC4 ectodomain treatments temporally protect cartilage damage in OA model and decrease ADAMTS-5 and MMP-13 expression while increasing TIMP3 expression in human OA chondrocytes. The presented study is very descriptive. Results are not strongly supportive of the conclusion. No significant scientific progress is made. The manuscript is very poorly written. Figure legends are not properly described and mixed with results. Actual figure # and figure # in the description in the result section are not matching. Duplicated paragraphs in the result section. Most bar graph labels are illegible, and most bar graphs are looked unprofessional. The manuscript is poorly organized and premature to publish.

Author Response

To Reviewer 2

The manuscript by Hattori, et al. entitled: “ Role of Syndecan-4 in Inhibiting Articular Cartilage 2 Degeneration in Osteoarthritis” (biomedicines-2492434) presents a study on the effects of SDC4 ectodomain treatments on articular cartilage degradation in surgically induced OA in mice and expression of the selected catabolic and anabolic factors in human OA chondrocytes in vitro. The authors found that SDC4 ectodomain treatments temporally protect cartilage damage in OA model and decrease ADAMTS-5 and MMP-13 expression while increasing TIMP3 expression in human OA chondrocytes. The presented study is very descriptive. Results are not strongly supportive of the conclusion. No significant scientific progress is made. The manuscript is very poorly written.

Thank you very much for kind review on our manuscript.

Figure legends are not properly described and mixed with results.

According to your kind comment, we have properly corrected figure legends so as not to be mixed with the results and the figure as follows.

Figure 1. Distribution of injected SDC4(SDC4: red) in the cartilage and synovium. Nuclei are labeled in blue. (Lines 252)

Figure 2. Histologic analysis of surgically-induced OA in the cartilage tissue of the knee joints of mice following treatment with SDC4 or phosphate-buffered saline (PBS) control; hematoxylin and eosin (H&E) and safranin-O staining.  (Lines 274-275)

Figure 3. (a) Mankin Score, (b) OARSI score. All parameters were taken as counts of the cartilage tissue. (Lines 279)

Figure 4. Histologic analysis of synovitis in the synovium tissue of the knee joints of mice after treatment with SDC4 or PBS control. H&E staining. (Lines 289-290)

Figure 5. Synovitis Scores in both groups at 2 and 4 weeks. All parameters were taken as counts of the synovium tissue. (Lines 294)

Figure 6. Immunohistochemical analysis of the cartilage tissue (Line 306)

Figure 7. (a) Percentage of ADAMTS-5–positive chondrocytes as determined by immunostaining. (b) Percentage of TIMP-3–positive chondrocytes as determined by imunostaining. All parameters were taken as counts of the cartilage tissue. (Lines 311-313)

Figure 8. Analysis of mRNA expression in cultured cells.  MMP-13: n=6, ADAMTS-4: n=8, ADAMTS-5: n=5, TIMP-3: n=9, bFGF: n=9, TGFb: n=9, and glyceralde-hyde-3-phosphate dehydrogenase (GAPDH): n=9 (Lines 324-326)

Actual figure # and figure # in the description in the result section are not matching.

According to your kind comment, we corrected the mismatching between actual figure # and figure # in the description in the result section as follows.

In contrast, the articular cartilage in the SDC4-treated group showed less proteoglycan loss. Articular lesions were assessed on a scale of 0–14 using the Mankin score (Fig. 3a), and on a scale of 0–6 using the OARSI grading (Fig. 3b). (Lines 262-264)

Low-grade synovitis occurred at 2 and 4 weeks in both groups, but improved by 6 weeks (Fig. 4). There were no significant differences in the average synovitis scores between the SDC4 and control groups at any time point (Fig. 5). (Lines 283-286)

The expression of ADAMTS-5 and TIMP-3 was examined in the cartilage of OA model mice in the SDC4 and control groups each week using immunohistochemical analysis (Fig. 6). (Lines 296-298)

No significant staining was observed with anti–MMP-13 antibody. In the SDC4 group at 8 weeks after treatment, the percentage of cells that stained positive for ADAMTS-5 was significantly lower than that in the control group (Fig. 7a), whereas the percentage of cells that stained positive for TIMP-3 was significantly higher than that in the control group (Fig. 7b) (Lines 298-302)

The function of SDC4 in regulating the expression of catabolic and anabolic factors in human OA chondrocytes was investigated using real-time PCR. (Fig. 8) (Lines 315-316)

Duplicated paragraphs in the result section. Most bar graph labels are illegible, and most bar graphs are looked unprofessional.

According to your kind comment, we modified the graph labels on each figure.

Reviewer 3 Report

The original research article assesses the role of syndecan-4 in inhibiting articular cartilage 2 degeneration in osteoarthritis. The topic is relevant and up-to-date, but certain deficiencies identified in both content and form need to be addressed based on the specific recommendations below:

Abstract- once abbreviated osteoarthritis (OA) only the abbreviated form will be used. Main text is treated separately from this point of view following the same rules.

The introduction should be significantly improved (as it goes directly and abruptly to Syndecan-4 without a prior presentation of the pathology) with data related to the prevalence, incidence and pathophysiology of OA, as well as its current management (non-pharmacological, pharmacological, rehabilitative) in order to identify current unmet needs that could be addressed by the proposed model. I suggest you check and consult: PMID: 35386619, PMID: 35454333 and PMID: 33815614.

L54- during tissue repair11 - please correct

The purpose of the paper in the last paragraph of the introduction should be approached from the perspective of describing the contribution to the field under review and the elements of scientific novelty presented.

Subsection 2.1 should be renamed because it is not a very scientific title 

L201-203 please remove unnecessary content

Figure x should be bolded as in the template provided by the journal.

L352-354 since it is consecutively presented information it is only necessary at the end of the bibliographic index [x].

The limitations presented should be discussed how they could be solved in future research directions showing how the results of this study can be used in practice at clinical level.

The conclusion section should be better detailed with the important aspects as it is too poorly detailed in relation to the complexity of the subject.

Author Response

To Reviewer 3

  • The original research article assesses the role of syndecan-4 in inhibiting articular cartilage 2 degeneration in osteoarthritis. The topic is relevant and up-to-date, but certain deficiencies identified in both content and form need to be addressed based on the specific recommendations below:

Thank you very much for kind review on our manuscript.

  • Abstract- once abbreviated osteoarthritis (OA) only the abbreviated form will be used. Main text is treated separately from this point of view following the same rules.

According to your kind comment, in the abstract, once abbreviated osteoarthritis (OA) we abbreviated it as osteoarthritis (OA) and used only that abbreviation thereafter. (Line 12-16)

Main text is treated separately from abstract following the same rules. (Lines 30,31)

  • The introduction should be significantly improved (as it goes directly and abruptly to Syndecan-4 without a prior presentation of the pathology) with data related to the prevalence, incidence and pathophysiology of OA, as well as its current management (non-pharmacological, pharmacological, rehabilitative) in order to identify current unmet needs that could be addressed by the proposed model. I suggest you check and consult: PMID: 35386619, PMID: 35454333 and PMID: 33815614.

According to your suggestion, we referred to the literature proposed and improved the introduction with date related to the prevalence, incidence and pathophysiology of OA, as well as its current management. (Lines 30-46)

  • L54- during tissue repair11 - please correct

According to your kind comment, we have made the following correction.

repair11→ repair [15] (Line 70)

  • The purpose of the paper in the last paragraph of the introduction should be approached from the perspective of describing the contribution to the field under review and the elements of scientific novelty presented.

According to your suggestion, we added the contribution to the field under review and the elements of scientific novelty presented at the end of the paragraph. We added the following sentence.

This study presents new findings that investigate the role of the SDC4 extracellular domain in articular cartilage. (Lines 79, 80)

  • Subsection 2.1 should be renamed because it is not a very scientific title

According to your suggestion, we renamed Subsection 2.1 as the follow.

2.1. Animals → 2.1. Experimental Animal Models (Line 82)

  • L201-203 please remove unnecessary content

According to your suggestion, we have deleted the sentence.(Lines 238-239)

  • Figure x should be bolded as in the template provided by the journal.

According to your kind comment, we corrected “Figure x” at each figure legends to bolded.

L352-354 since it is consecutively presented information it is only necessary at the end of the bibliographic index [x].

According to your kind comment, we described “[27]“at the end of bibliographic index. (Lines 360-362)

  • The limitations presented should be discussed how they could be solved in future research directions showing how the results of this study can be used in practice at clinical level.

According to your suggestion, we added to the Limitation on the future potential and direction for clinical application as follows.

Elucidating the signaling pathway through which the extracellular domain of SDC4 inhibits articular cartilage degeneration and exerts other effects of SDC4 on articular cartilage may help in the development of therapies that inhibit OA progression. (Lines 406-409)

  • The conclusion section should be better detailed with the important aspects as it is too poorly detailed in relation to the complexity of the subject.

According to your suggestion, we better detailed with the important aspects in the conclusion section as follows.

In the in vivo study, SDC4 inhibited cartilage degeneration of OA model mice at 6 and 8 weeks without synovitis. In OA model mice, SDC4 treatment decreased ADAMTS-5 expression and increased TIMP-3 expression in chondrocytes after 8 weeks. In the in vitro study, SDC4 treatment upregulated the expression of TIMP-3 and downregulated the expression of MMP-13 and ADAMTS-5 in chondrocytes. (Line 412-416)

Round 2

Reviewer 3 Report

The authors have significantly improved the manuscript based on the suggestions received.

Author Response

To Academic Editor Notes

Thank you very much for kind review on our manuscript.

  • The introduction on OA needs to be improved. At line 38, it should be clearly added that OA is a whole joint disease involving all joint tissues: cartilage, meniscus, synovial membrane, infrapatellar fat pad and subchondral bone.

According to your suggestion, we added the following sentences in introduction.

OA is a whole joint disease involving all joint tissues: cartilage, meniscus, synovial membrane, infrapatellar fat pad and subchondral bone. (Lines 37-39)

  • lines 39-43: references should be added.

According to your kind comment, we added the reference as follows.

Degradative enzymes, also known as matrix metalloproteinases (MMPs), are upregulated in OA, triggering an imbalance that leads to the loss of proteoglycans and collagen. During OA development, chondrocytes tend to increase proteoglycan synthesis and produce tissue inhibitory factors for MMPs (TIMPs) to balance degradation, with restorative actions being inadequate to counteract these changes [5]. This imbalance leads to a decrease in proteoglycan content despite an increase in synthesis, higher water content, disrupted collagen structure, and reduced articular cartilage elasticity. These changes cause wear on the joint surfaces [6]. (Lines42-49)

  • The authors analzyed also synovial membrane. Thus, an appropriate introduction on this tissues should be added. Moreover, the infrapatellar fat pad needs to be mentioned in the discussion/introduction as it is closed to synovial membrane (adipocytes are also visible in the images).

According to your suggestion, we added the following sentences in introduction.

Synovitis is an important feature in OA process and is defined as inflammation of the synovium [21]. It may manifest itself phenotypically as thickening of the synovial membrane or indirectly as joint effusion as the result of synovial activation [22]. One study demonstrated a positive correlation between the severity of synovitis and the degree of progression of cartilage lesions over time [23], suggesting that synovitis in OA predisposes to further structural progression [24, 25]. Infrapatellar fat pad (IFP) is an intracapsullar and extrasynovial adipose tissue structure in the knee joint that is closely associated with synovitis [26]. It is suggested that the IFP could be an important player in OA [27]. IFP could have both protective and disease-enhancing effects in OA [26]. In light of previous reports that identified shed SDC4 in the joint fluid of patients with OA [19] and that SDC4 is associated with OA progression [15], the possibility that Shed SDC4 induces synovitis must also be considered. (Lines 76-87)

  • line 97: it should be specified (rhSDC4, Glu19-Glu145; R&D Systems, Minneapolis, MN, USA, #2918-SD).

According to your kind comment, we made the following correction.

(rhSDC4; R&D Systems, Minneapolis, MN, USA, #2918-SD)

(rhSDC4, Glu19-Glu145; R&D Systems, Minneapolis, MN, USA, #2918-SD)

(Line 110)

  • Figure 1: Synovial tissure needs to be fixed.

According to your kind comment, we made the following correction.

 “ Synovial tissure “ → “ Synovium ”  (In Figure 1)
